# Prognostic ^18^F-FDG Radiomic Features in Advanced High-Grade Serous Ovarian Cancer

**DOI:** 10.3390/diagnostics13223394

**Published:** 2023-11-07

**Authors:** Daniela Travaglio Morales, Carlos Huerga Cabrerizo, Itsaso Losantos García, Mónica Coronado Poggio, José Manuel Cordero García, Elena López Llobet, Domenico Monachello Araujo, Sebastián Rizkallal Monzón, Luis Domínguez Gadea

**Affiliations:** 1Nuclear Medicine Department, La Paz University Hospital, 28046 Madrid, Spain; 2Nuclear Medicine Department, Halle University Hospital, 06120 Halle, Germany; 3Department of Medical Physics and Radiation Protection, La Paz University Hospital, 28046 Madrid, Spain; 4Biostatistics Department, La Paz University Hospital, 28046 Madrid, Spain

**Keywords:** radiomics, PET, texture features, heterogeneity, ovarian cancer, prognosis

## Abstract

High-grade serous ovarian cancer (HGSOC) is an aggressive disease with different clinical outcomes and poor prognosis. This could be due to tumor heterogeneity. The 18F-FDG PET radiomic parameters permit addressing tumor heterogeneity. Nevertheless, this has been not well studied in ovarian cancer. The aim of our work was to assess the prognostic value of pretreatment 18F-FDG PET radiomic features in patients with HGSOC. A review of 36 patients diagnosed with advanced HGSOC between 2016 and 2020 in our center was performed. Radiomic features were obtained from pretreatment ^18^F-FDGPET. Disease-free survival (DFS) and overall survival (OS) were calculated. Optimal cutoff values with receiver operating characteristic curve/median values were used. A correlation between radiomic features and DFS/OS was made. The mean DFS was 19.6 months and OS was 37.1 months. Total Lesion Glycolysis (TLG), GLSZM_ Zone Size Non-Uniformity (GLSZM_ZSNU), and GLRLM_Run Length Non-Uniformity (GLRLM_RLNU) were significantly associated with DFS. The survival-curves analysis showed a significant difference of DSF in patients with GLRLM_RLNU > 7388.3 versus patients with lower values (19.7 months vs. 31.7 months, *p* = 0.035), maintaining signification in the multivariate analysis (*p =* 0.048). Moreover, Intensity-based Kurtosis was associated with OS (*p* = 0.027). Pretreatment ^18^F-FDG PET radiomic features GLRLM_RLNU, GLSZM_ZSNU, and Kurtosis may have prognostic value in patients with advanced HGSOC.

## 1. Introduction

Ovarian cancer (OC) is the third main cause of gynecologic cancer correlated death and is the fourth most commonly diagnosed gynecologic cancer in the world [1].

Clinically, high-grade serous ovarian cancer (HGSOC) has an aggressive pattern and most of the cases are diagnosed in advanced stages due to the lack of symptoms in early stages. The prognosis of advanced HGSOC is poor, representing 70–80% of deaths from all forms of ovarian cancer [2]. 

Patients with a histologic diagnosis of HGSOC have different chemotherapy responses and clinical outcomes [3]. In fact, almost 80% of women with OC in advanced stages respond to first-line chemotherapy relapse [4], therefore, it is necessary to identify patients at high risk of relapse and investigate different maintenance treatments to reduce recurrence. Previous researchers have proposed that this behavior is due to cellular genomic diversity [5,6] and denominate heterogeneity. This diverseness can be inter-tumoral when variations are found between the same type of tumors of different patients. Further, they are intra-tumoral when the distinct cellular populations with specific phenotypic features are in one single tumor, inter-metastatic when different cell populations are found in diverse lesions (primary tumor and metastasis), and intra-metastatic heterogeneity when differences are found in one single metastatic lesion [7]. Potential therapeutic targets have been identified in the genomic heterogeneity of the primary tumor. However, fewer advances have been made in peritoneal implants due to their localization and size making it difficult to obtain a biopsy [8,9]. For this reason, it is essential to develop methods for studying heterogeneity characteristics in vivo. On this basis, imaging techniques such as magnetic resonance (MR), computed tomography (CT), and positron emission tomography (PET) have become important to assess whole tumor characteristics. Specifically, PET imaging has been increasingly considered as an optimal instrument to study, describe, and categorize the biology of tumors at the macroscopic scale [10,11,12]. Quantification of PET images can deliver information about tumor heterogeneity [13].

Radiomic is a method that performs a quantitative assessment of visually imperceptible imaging parameters. A large amount of information is extracted and analyzed from Volumes of Interest (VOI) placed principally on solid tumors. First-order statistics (e.g., mean, median) and high-order parameters (texture features) are obtained. The texture features evaluate patterns and interrelationships between voxels inside the VOI. This method may contribute to the goal of individualized medicine and tumor evaluation [14].

There is a limited amount of data on the role of PET radiomic features in gynecological tumors. Nakajo et al. have found a predictive value of pretreatment Flourine-18 fluorodeoxyglucose positron emission tomography ([18F]-FDG PET) texture features in patients with endometrial cancer [15]. Regarding ovarian cancer, clinical factors such as age and staging or residual tumor after cytoreduction are traditionally used as prognostic factors. However, studies have proved that molecular tumor characteristics, such as the expression of progesterone receptors and elevated levels of human epidermal growth factor receptor 2 can predict survival in ovarian cancer [16]. Wang et al. [17] al have observed an association between pretreatment PET radiomic features integrated with clinical factors with progression-free survival in patients with HSOC in China.

However, the correlation between radiomic PET signatures and clinical outcomes in HGSOC is not yet confirmed. Therefore, in this study, we investigate the relationship between pre-treatment radiomic parameters and clinical follow-up in patients with advanced serous high-grade ovarian cancer, in order to evaluate the prognostic value of these parameters.

## 2. Materials and Methods 

### 2.1. Patients

We retrospectively reviewed patients diagnosed and treated for advanced, high-grade serous ovarian cancer (International Federation of Gynecology and Obstetrics (FIGO) staging III and IV) in our institution, who underwent a PET/TC imaging prior to initial therapy between 2016 and 2020, followed by surgery or neoadjuvant therapy. Patients were excluded if (1) they did have other ovarian cancer histology than high-grade serous ovarian cancer (2) the PET/CT was not performed for pre-treatment initial staging, (3) they had initial FIGO stages (I–II) of HGSOC, (4) they received any kind of treatment before PET/CT, (5) underwent adnexectomy for biopsy before PET/CT, or (6) PET/CT acquisition was not carried out in our center. The age, date of diagnosis, histological type of tumor, FIGO, treatment data, PET/CT date, date of relapse, actual status, and date of death/last consultation were collected. All the patients were treated and followed, clinically and radiologically, according to the institutional tumor board. Clinical data were extracted from the Dedalus-HCIS Healthcare Information System Version 1. This study was approved by the ethics committee of our institution. 

### 2.2. PET/CT Imaging Technique 

PET/CT studies were acquired in a GE Discovery LS scanner (GE Healthcare, Milwaukee, Brookfield, WI, USA). All patients fasted for at least 4 h prior to the acquisition. The imaging was performed 60 min after intravenous injection (IV) of a standard dose, 370 MBq, of ^18^F-FDG. Patients lay supine with their arms above the head. After the topogram, low-dose CT without IV contrast was acquired from the skull base to the proximal thighs, followed by 2D PET acquisition (4 min per bed). The images were reconstructed using ordered subset expectation maximization with CT-based attenuation correction without point spread function (PSF) correction. The PET images were reconstructed in a 128 × 128 matrix with a voxel size of 3.906 mm × 3.906 mm × 4.250 mm.

Imaging analysis was performed by two different experienced nuclear medicine physicians, separately, blinded to the clinical outcomes at interpretation. Disagreements about the interpretation were resolved through consensus. The PET VCAR application from the AW Volume Share 7 version AW4.7 of the Advantage Workstation (GE Healthcare Medical Systems, San Francisco, CA, USA) was used. First, pathological findings in relation to ovarian cancer were visually identified. Afterward, we individually defined a maximum Standardized Uptake Value (SUVmax) threshold that ensured the preselection of all pathologic lesions that met Positron Emission Tomography Response Criteria in Solid Tumors (PERCIST criteria). Then, radiomic parameters were obtained by drawing VOIs semiautomatically, enclosing the primary tumor and metastatic lesions in the transaxial, sagittal, and coronal projections from PET images. The VOIs were made using boundaries of voxels with a threshold of 41% of SUVmax to define the tumor volumes. Physiological uptake of the heart, liver, kidneys, ureters, and bladder, as well as inflammatory lesions or other non-tumoral or primary neoplasms, were excluded manually. The software classified the contoured lesions as Target or non-Target according to SUVmax normalized by lean body mass (SULmax), being the Target lesions the ones with maximum SULmax. Total lesion parameters were calculated by summing the parameters of target and non-target lesions.

### 2.3. Radiomic Features Extraction

Radiomic feature extraction from PET images was performed with LIFEx software (version 7.2.0) [18]. A total of 85 radiomic features were extracted. These are grouped into 12 morphological (shape-based), 17 intensity-based (extracted from the histogram and local intensity) and 56 texture features grouped in four grey-level matrixes: Gray-Level Cooccurrence Matrix (GLCM), Gray level Run-Length Matrix (GLRLM), Neighbourhood grey tone difference (NGTDM) and Grey-Level Zone Length Matrix (GLZLM) also named Grey Level Size Zone Matrix (GLSZM). LIFEx software complies with the Image Biomarker Standardization Initiative (IBSI) as was demonstrated by Fornacon-Wood et al. [19]. The primary aim of this initiative is to improve the reproducibility of quantitative image analyses. 

The following extraction parameters were applied: no scaling (no spatial resampling), VOI isotropy was assumed. The extraction of the parent matrixes from the VOI was carried out using a 3D analysis in 13 directions using a distance of 1 voxel except for GZLM computed directly in 3 dimensions.

Two different discretized gray levels were applied. Fixed bin number of 64 levels, between 0 and 20 standardized uptake value (SUV) units, as is recommended by Orlhac et al. [20], and relative resampling with gray levels scaled between minimum and maximum values of the VOI with a bin size of 0.5 SUV.

The dimensionality reduction was performed using pairwise correlation between the features of each group or parent matrix with a correlation threshold value of 0.6. 

After removing redundant features, 23 of them remain, of which 4 are shape-based, 5 are intensity-based, and 14 are texture features. Table 1 shows the selected characteristics. 

### 2.4. Clinical End Points and Follow-Up 

Data on relapse, death, and other clinical information were collected from medical records. Clinical endpoints were Disease Free Survival (DFS), defined as the time between diagnosis and date of relapse/progress documented by clinical symptoms, Ca 125 levels and imaging studies, and Overall Survival (OS), defined as the time between diagnosis and death related to primary ovarian cancer.

### 2.5. Statistical Analysis

Qualitative data (clinical data) were described as absolute frequencies and percentages, and quantitative data as mean and standard deviation or median and interquartile range, depending on the distribution of these data.

The normality of continuous variables was studied using the Kolmogorov-Smirnov test.

The relationship between quantitative variables was studied using Pearson’s correlation or its nonparametric equivalent Spearman. 

Optimal cutoff points were obtained (ROC analysis) of those variables with statistical significance, which classifies the population between patients who have an event (relapse or death) or not. Survival curves were drawn using the Kaplan–Meier method. Log-rank tests were used to compare survival functions by groups. A subgroup analysis categorized by median values and type of treatment were made. The risks associated with each variable were studied by the Cox regression analysis (univariate and multivariate analysis).

All statistical tests were considered bilateral and as significant values, those *p* ≤ 0.05. The data were analyzed with the SAS 9.3 statistical program (SAS Institute, Cary, NC, USA). 

## 3. Results 

### 3.1. Patient Characteristics 

Initially, studies of 260 consecutive patients with ovarian cancer were obtained between January 2012 and September 2020 in our institution. Twenty-six patients were excluded as they had other histology results different from high-grade serous ovarian cancer. A total of 137 patients were eliminated because PET/CT was performed for treatment monitoring or relapse diagnosis. Seven patients were excluded since they had initial FIGO stages (I–II). Twenty-four patients were removed as they underwent an exploratory laparoscopy with some cytoreduction or received some cycle of chemotherapy and 4 patients had an adnexectomy as a biopsy before PET/CT acquisition. Finally, 62 patients with high-grade serous ovarian cancer, advanced stage (III–IV FIGO staging), and basal PET/CT were collected. Six patients had no follow-up. Images were not available for analysis on 3 patients. Subsequently, 53 patients with high-grade serous ovarian cancer, advanced stage (III–IV FIGO staging), and basal PET/TC were candidates for segmentation. Because of low tumor-to-background uptake ratios, metabolic parameters could not be calculated in 4 patients; moreover, the PET/CT acquisition of 13 patients was made outside our institution. Finally, 36 patients were eligible for analysis, with a mean age of 60 years old (range 42–84). A flowchart showing the exclusion criteria and the process of patient selection is presented in Figure 1. The patient’s characteristics are indicated in Table 2. The mean follow-up was 31.19 months. Relapse and exitus occurred in 72.2% and 41.7% of patients, respectively. The mean values of DFS and OS were 19.6 and 37.1 months, respectively.

### 3.2. Radiomic Parameters and DFS Association

The descriptive statistical values obtained for the radiomic characteristics are summarized in Table 3. 

A statistically significant relation was found between Intensity-based Total Lesion Glycolysis (TLG) and GLSZM_ Zone Size Non-Uniformity (GLSZM_ZSNU) with DFS (*p* = 0.041 and *p* = 0.030, respectively). In the survival analysis categorized by ROC cut-off value, patients with GLSZM_ZSNU < 1103.9 showed a greater risk of relapse than those with GLSZM_ZSNU > 1103.9, although this relation was not statistically significant (*p*: 0.206). Moreover, the association between subgroups (categorized by median) and DFS was not statistically significant.

GLRLM_Run Length Non-Uniformity (GLRLM_RLNU) had an association with DFS although not statistically significative (*p* = 0.060). This association became significant when different gray levels resampling (in relative mode) was used (*p* = 0.045). In the survival analysis using ROC cut-off value, we found that patients with GLRLM_RLNU value < 7388.3 had more favorable DFS than those with GLRLM_RLNU > 7388.3 (*p* = 0.035, HR = 0.402 with a 95% CI, 0.167–0.965). DFS curves considering GLRLM_RLNU ROC cut-off value are shown in Figure 2. 

### 3.3. Radiomic Parameters and OS Association

We found a significant association between Intensity-based Kurtosis and OS (*p* = 0.027). For each unit of incremented Kurtosis, the risk of death increased by 4.8%. However, no significant correlation was found in the subgroup analysis. OS curves considering Kurtosis are shown in Figure 3. Patients with Intensity-based Kurtosis value < 1.8 had a better OS than those with Intensity-based Kurtosis > 1.8, although not statistically significant (*p* = 0.053). 

Survival values considering GLRLM_RLNU, GLSZM_ZSNU, and Kurtosis are shown in Table 4.

### 3.4. Type of Treatment and DFS/OS Association

No statistic significant association was found between type of treatment and DFS (*p* = 0.09). An OS analysis was not possible because of the lack of events, however, patients with only chemotherapy had worse survival. Then, we analyzed the patients with neoadjuvant chemotherapy + interval debulking surgery and primary cytoreductive surgery + adjuvant chemotherapy together and compared them with the OS of chemotherapy patients. A significant association between these groups was found, having a worse survival the patients with only chemotherapy (*p* = 0.005 HR: 0.27 with a 95% CI, 0.968–0.700). Receiving a primary cytoreductive surgery or interval debulking reduces the risk of death in 78.3% of patients.

### 3.5. Multivariate Analysis: Radiomic Features (GLRLM_RLNU, GLSZM_ZSNU, and Kurtosis), Type of Treatment, and DFS/OS

The association between ROC cut-off values of GLRLM_RLNU with gray levels resampling and DFS maintain their significance (*p* = 0.048 HR = 2.35 with a 95% CI, 1.070–5.735). Patients with GLRLM_RLNU value > 7388.3 have 2.35 times more risk of relapse. 

GLSZM_ZSNU had an association with DFS, although not statistically significant (*p* = 0.057). No other significant associations were found. 

## 4. Discussion

In recent years, several researchers have evaluated the use of ^18^F-FDG PET radiomic parameters in the oncologic field, proven their impact in diagnosis, as a prognostic factor, and even their role in predictive models [15,21,22,23,24,25,26] contributing to the selection of the appropriate treatment and follow up for each patient. In ovarian cancer, some publications have studied radiomic features in CT or MR and their correlation with prognosis [26,27,28,29,30,31], although PET radiomic parameters are relatively less studied in this type of cancer [17]. The aim of our work was to study the relationship between radiomic parameters obtained in pre-treatment ^18^F-FDG PET/CT and clinical outcomes in patients with advanced HSOC.

In line with other publications [32,33,34], heterogeneity was an important prognostic factor in our cohort. We found that GLSZM_ZSNU has a significant association with DFS and GLRLM_RLNU had a tendency towards this correlation: patients with higher values of heterogeneity have a worse DFS. Moreover, GLRLM_RLNU with relative gray-level resampling was an independent prognostic factor for relapse. GLRLM_RLNU reflects the spatial distribution of runs of consecutive pixels with different gray levels in one or more directions [14]. When runs are equally distributed, the value of the feature is low. Otherwise, GLSZM_ZSNU counts the number of zones linked voxels and the distribution of zone counts over different zone sizes. When zone counts are equally distributed, the value of the feature is low [35]. Similar to our results, other authors [15,36,37,38] have found the prognostic significance of GLRLM_RLNU and GLSZM_ZSNU [39] and even a correlation with immunohistochemical factors [37] in other types of tumors. Nakajo et al. [15] have studied the role of pretreatment PET radiomic features in endometrial cancer as predictive factors. They have found that coarseness NGLDM, GLRLM_RLNU, GLZLM_ZLNU, and GLZLM_RLNU have a significative association with DFS. These results are in line with our data. NGLDM, GLSZM (or GLZLM), and GLDZM are structured similarly to GLRLM, therefore their features definitions are based on the definitions of GLRLM features [35]. Hence, GLRLM features and GLRLM-based features may have predictive value in gynecological cancers. There is little data regarding the predictive capacity of PET radiomic features in ovarian cancer. Wang et al. [17] have studied a predictive model using pretreatment PET and CT radiomic signatures integrating clinical and metabolic factors to prognosticate DFS in 261 patients with advanced high-grade serous ovarian cancer in China. They found that PET radiomic signature can improve diagnostic accuracy compared to clinical factors alone or combined with radiomic signatures in CT. Furthermore, it can predict DFS in this type of patient. They used 10 radiomic features selected by the least absolute shrinkage and selection operator method (LASSO) to construct a radiomic signature. In accordance with our work, one of these features was GLSZM_ZSNU but normalized. This could be due to differences between samples and/or calculation softwares. Moreover, others of the features chosen were GLRLM features and based features. In contrast with our study, they did not find any correlation between TLG and DFS. This may be due to differences in the definition of follow-up points, the number of patients, and the choice of lesion for obtaining radiomic parameters. In accordance with our findings, some authors have established this correlation. Chung et al. [40] and Lee J et al. [41] have found that preoperative ^18^F-FDG PET TLG and metabolic tumor volume (MTV) are associated with DFS in populations with ovarian cancer with different histologies and stages. Our work did not find any relation with MTV, which may be due to the small sample. Liu et al. [42] have demonstrated in patients with high-grade serous ovarian cancer that high values of preoperative PET TLG and MTV of the primary tumor are associated with better chemosensitivity and better OS. We did not find any association between TLG and OS. Our findings are not completely comparable because we use different criteria for choosing the VOI to be studied. 

We also found an association between OS and Kurtosis. Higher Kurtosis values tended to have shorter OS. This parameter has been associated with OS as a prognostic factor in other tumor histologies as pancreatic cancer [43], prostate cancer [44], or Gastric Diffuse Large B-Cell Lymphoma [45]. 

Our study has some limitations. As a retrospective study, it probably has a selection bias. We only included patients with pretreatment ^18^F-FDG PET/CT performed in one scanner and excluded patients with low ratio tumor background (in order to achieve the segmentation process). This could possibly misrepresent the entire population. Furthermore, only a small number of patients were included. Consequently, we were unable to implement a predictive model nor external validation. Our sample was treated differently according to the guidelines of HSOC Therapy. In fact, the patients who received only chemotherapy had a worse OS in our sample. This might lead to bias; nonetheless, this could show our results as parameters with potential implementation in daily clinical practice. Moreover, no statistically significant radiomic feature differences were found between subgroups categorized by type of treatment.

Semiautomatic segmentation requires a careful delimitation of abnormal hypermetabolic areas, differentiating them from normal uptake of physiological processes. In our segmentation method, we avoided including lesions smaller than 1 cm^3^, unless they had a considerable uptake. Furthermore, we did not add lesions manually except for the most obvious ones. This could be a limitation in this type of disease as its peritoneal spreading, resulting in numerous small lesions around physiological uptake. Thus, it is important to establish further automatic segmentation methods more suitable for this type of disease spread. The designation of target lesions through high uptake could be a possibility. 

Although only one scanner was used in our study, a possible limitation may be the reproducibility of the radiomic parameters. A review from Traverso et al. [46] compiled different studies of radiomic features, reproducibility, and repeatability. They observed that texture features in PET studies showed greater sensitivity to segmentation differences and reconstruction algorithms. Shape metrics reproducibility was mainly affected by the reconstruction algorithm. First-order PET features have normally more robustness [46]. Although there is some disagreement over the best percentage of boundaries of voxels as a segmentation method for delineating lesions, some authors have demonstrated that there is no significant impact between them to obtain radiomic features [47]. Hence, there is a need for a consensus on imaging reconstruction, radiomic features extraction, and analysis, to find a methodology that makes the use of radiomics more reliable and reproducible. As future perspectives, a multicenter study exploring radiomic features in ovarian cancer could contribute to the development of this field, reaffirming and expanding our findings, with a larger sample and different scanners, seeking standardization, reproducibility, and clinical practice implementation. We believe that more studies exploring GLRLM features and GLRLM-based features are noteworthy, since these features may have an important value in gynecologic cancer. The investigation of first-order features such as MTV and TLG is also needed as they are easier to implement in clinical practice. Moreover, a comparison between PET radiomic parameters and other imaging methods, such as CT or MRI, should be conducted since radiomic parameters of morphologic imaging could complement the metabolic information. A promising technique is PET/MRI, but unfortunately, it has limited availability. 

In conclusion, we have studied the prognostic value of radiomic features in a very homogeneous series of patients, with the same histological subtype and clinical stage in pretreatment PET/CT. This report is noteworthy; to our knowledge, it is the first study that has identified the prognostic value of specific PET radiomic features in patients with HSOC at advanced stages. Moreover, it is the first study performed in a European population. In this context, radiomic features, especially heterogeneity characteristics, may be useful to identify a subgroup of patients with poor prognosis. Our results reflect the need for further investigations on radiomic parameters in ovarian cancer. 

## Figures and Tables

**Figure 1 diagnostics-13-03394-f001:**
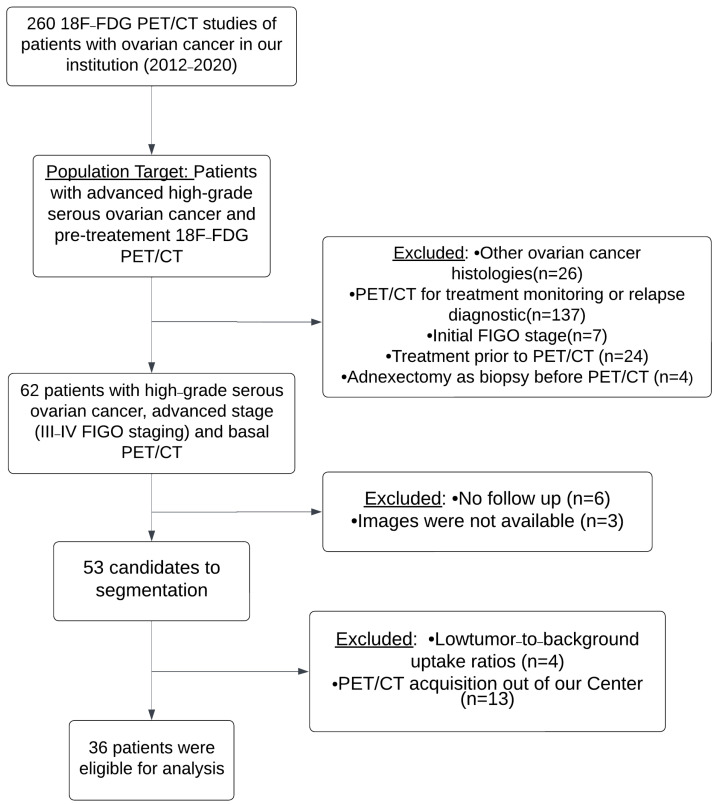
Flowchart showing the patient selection process.

**Figure 2 diagnostics-13-03394-f002:**
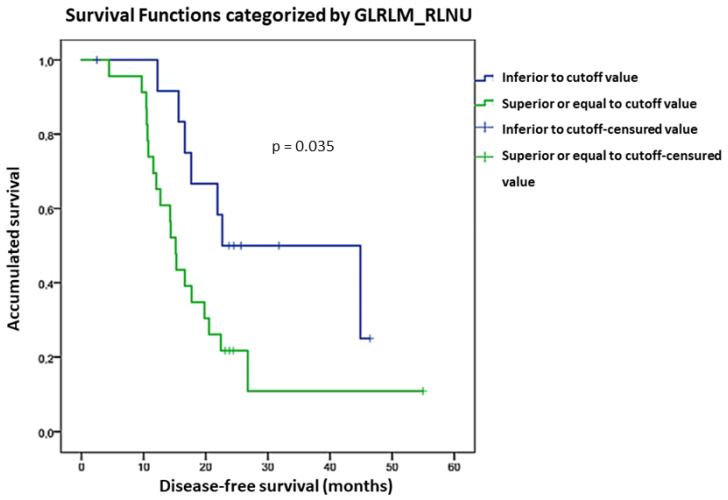
DFS curves considering GLRLM_RLNU ROC cut-off.

**Figure 3 diagnostics-13-03394-f003:**
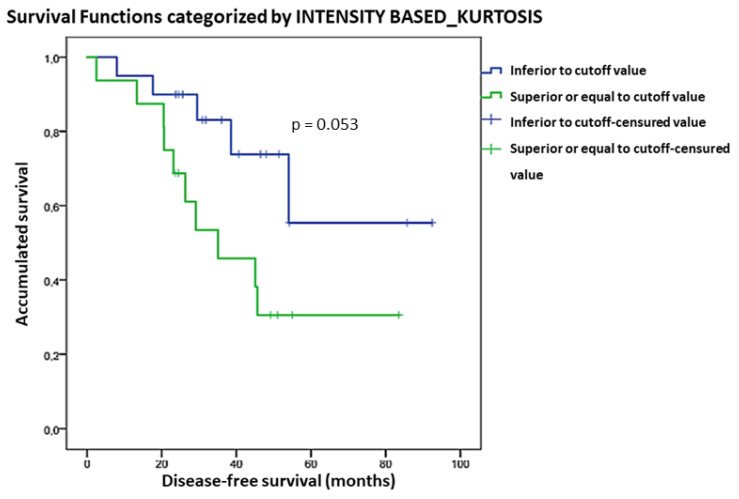
Kaplan-Meier OS curves considering the Kurtosis cut-off value.

**Table 1 diagnostics-13-03394-t001:** A list of 23 PET-based radiomic features selected.

Matrix	Index
First order features	
Morphological	Approximate Volume
	MORPHOLOGICAL_Compacity
	MORPHOLOGICAL_Compactness2
	MORPHOLOGICAL_Centre OF Mass Shift
Intensity-based	Total Lesion Glycolysis
	Variance
	Kurtosis
	Minimum Grey Level
	Histogram Uniformity
Higher order features	
Gray-Level Cooccurrence Matrix (GLCM)	GLCM_Joint Maximum
	GLCM_Inverse Difference Moment
	GLCM_Inverse Variance
	GLCM_Correlation
	GLCM_Cluster Tendency
	GLCM_Cluster Shade
Neighborhood grey tone difference (NGTDM)	Coarseness
	Contrast
	Busyness
Gray level Run-Length Matrix (GLRLM)	Long Runs Emphasis
	Run Length Non Uniformity
Grey-Level Zone Length Matrix (GLZLM) or Grey-Level Size Zone Matrix (GLSZM)	Large Zone High Grey Level Emphasis
	Zone Size Non Uniformity
	Normalised Zone Size Non Uniformity

**Table 2 diagnostics-13-03394-t002:** Patient Characteristics (*n* = 36).

Characteristic	Patients
Total patients	36
Mean age, years (range)	60 (42–84)
FIGO stage	
III	12 (33.3%)
IV	24 (66.7%)
Histology	
High-grade serous carcinoma	36
Type of treatment	
Neoadjuvant chemotherapy + interval debulking surgery	20 (55.6%)
Primary cytoreductive surgery + adjuvant chemotherapy	8 (22.2%)
Chemotherapy only	8 (22.2%)
Chemotherapy	
Carboplatin + paclitaxel with Bevacizumab + Bevacizumab as maintenance	6 (16.6%)
Carboplatin + paclitaxel + Bevacizumab as maintenance	13 (36.1%)
Carboplatin + paclitaxel without Bevacizumab	3 (8.3%)
Carboplatin + paclitaxel with Bevacizumab without maintenance	14 (38.9%)
PARPi	1 (2.7%)
Mean Follow-up months	31.19
Mean DFS, months	19.6 ± 11
Mean OS, months	37.1 ± 20.3

**Table 3 diagnostics-13-03394-t003:** Descriptive statistical analysis of PET parameters.

*n* = 36	MTV	TLG	_Kurtosis	GLSZM_ZSNU	GLRLM_RLNU
Mean	1025.7	3828.1	4.3	1275.7	11,100.4
Median	756.8	2885.5	1.3	1059.2	8017.6
Max	4016.2	13,023.0	58.7	5265.0	42,719.5
Min	19.2	154.0	−0.8	60.0	194.9
SD	973.8	3383.4	10.1	1225.7	10,107.9

MTV: Metabolic tumor volume, TLG: Total Lesion Glycolysis, _Kurtosis: Intensity based Kurtosis, GLRLM_RLNU: Gray level Run Length Matrix Run Length Non Uniformity, GLSZM_ZSNU: Grey Level Size Zone Matrix Zone Size Non Uniformity.

**Table 4 diagnostics-13-03394-t004:** Survival analysis.

Feature	ROC Cut Off	Se	Sp	Mean DFS/OS (Months)		*p*	HR (CI 95%)	*p*
				Group Superior to Cut-Off	Group Inferior to Cut-Off			
**DSF analysis**								
GLRLM_RLNU	7388.3	0.73	0.60	19.7	31.7	**0.035** *	0.402	**0.041** *
GLSZM_ZSNU	1103.9	0.50	0.60	21.6	26.9	0.206	-	-
**OS analysis**								
_Kurtosis	1.8	0.66	0.71	44.3	68	0.053	-	-

* Indicates significance with a *p*-value of <0.05 (shown in bold). Se: Sensibility, Sp: Specificity, GLRLM_RLNU: Gray level Run-Length Matrix Run Length Non Uniformity, GLSZM_ZSNU: Grey-Level Size Zone Matrix Zone Size Non-Uniformity, _Kurtosis: Intensity-based Kurtosis.

## Data Availability

The datasets analyzed and generated for this study can be found in the Radiomics data base 2.0.xlsx and Radiomics statistic results 2.0.pdf files in Appendix A.

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
