# Peer review of "Prognostic 18F-FDG Radiomic Features in Advanced High-Grade Serous Ovarian Cancer"

_diagnostics, 2023, doi:10.3390/diagnostics13223394_

Round 1

Reviewer 1 Report

Comments and Suggestions for Authors

This is a well-written article on the prognostic role of 18F-FDG radiomic features in advanced high grade serous ovarian cancer. My only suggestion is to end with a section on potential future directions. Given the findings, what would be the next steps? Are there any other radiomic parameters worth exploring? What about other imaging modalities?

Comments on the Quality of English Language

Well used grammar.

Author Response

Reviewer 1:

 My only suggestion is to end with a section on potential future directions. Given the findings, what would be the next steps? Are there any other radiomic parameters worth exploring? What about other imaging modalities?

We believe that the next step should be a multicenter study with a larger sample and different scanners, seeking reproducibility and implementation in daily clinical practice. Such a study may reaffirm our results, establishing a field for the implementation of radiomics. According to our work and available references, GLRLM features and GLRLM-based features should be explored for validation and practical use. Moreover, first-order parameters such as MTV and TLG should be studied in larger samples. An interesting study would be the comparison between radiomic features obtained from morphological images (CT and MRI) and PET radiomics, as they could provide more information for tumor characterization. We include a paragraph at the end of the discussion with these ideas.

Reviewer 2 Report

Comments and Suggestions for Authors

I think this report is meaningful, because these results showed the relationship between PET findings and clinical outcomes. And if the sample size become larger, the ROC cut-off value for patients with GLSZM_ZSNU can show the significant relationship. So, I have no request.

Author Response

Thank you very much for the positive comments!!

Reviewer 3 Report

Comments and Suggestions for Authors

Daniela et al. have shown the 18FDG-PET features exhibit prognostic values in patients with HGSOC by reviewing data from 36 patients. This study may have clinical implications and hold a specific interest to clinicians. I have a few questions and hope the authors can address them.

1.      The first drawback of this study is the small sample size, which could weaken the conclusions. Nevertheless, it is still worth investigating the clinical significance of PETCT. I suggest the authors run subgroup analyses based on different treatments. The point is that NACT or debulking surgery may influence the patient’s survival.

2.      Can the author provide the chemotherapy details of each patient? Have they received bevacizumab or PARPi?

3.      What is the gene status of those patients? Is there any relationship between the gene status and PETCT features?

4.       The author should provide more information about the GLZLM-ZLNU.

Author Response

Reviewer 3:

The first drawback of this study is the small sample size, which could weaken the conclusions. Nevertheless, it is still worth investigating the clinical significance of PETCT. I suggest the authors run subgroup analyses based on different treatments. The point is that NACT or debulking surgery may influence the patient’s survival.

We analysed the influence of the different types of treatment and the patients outcome. Kaplan Meier Curves were compared between treatments subgroups. No statistic significant association was found between type of treatment and disease free survival (p=0,09). A OS analysis was not possible because of the lack of events, however patients with only chemotherapy apparently had a worse survival. Then, we analysed the pacientes with NACT and Debulking surgery together and compared with the OS of only chemotherapy patients. A significative diference between these groups were found, having a worse survival the patients with only Chemotherapy (p=0,005 HR:0.27). No significant difference between radiomic features values and type of treatment were found. A multivariate analysis considering radiomic features, type of treatment and DFS/OS was done. The association between ROC cut-off values of GLRLM_RLNU with gray levels resampling and DFS maintain their significance (p=0.048 HR=2.35 with a 95% CI, 1.070-5.735). Patients with GLRLM_RLNU value >7388.3 have 2.35 times more risk of relapse. Being a independent prognostic factor for relapse.

GLSZM_ZSNU had association with DFS although not statistically significant (p=0.057).  No other significant associations were found. Althought patients with chemotherapy only have worse OS, radiomic parameters have also a influence on the patient outcome, having GLRLM_RLNU the strongest association with DFS.

We have added the new analysis to our results.

Can the author provide the chemotherapy details of each patient? Have they received bevacizumab or PARPi?

From 36 patients only one received a PARPi.

The chemotherapy details are:

Patients with chemotherapy carboplatin + paclitaxel with Bevacizumab + Bevacizumab as mantainance: 6 (16.6%)

Patients with chemotherapy carboplatin + paclitaxel + Bevacizumab as mantainance: 13 (36,1%)

Patients with chemotherapy carboplatin + paclitaxel without Bevacizumab: 3 (8.3%)

Patients with chemotherapy carboplatin + paclitaxel with Bevacizumab without mantainance: 14 (38.9%)

This information is added on the table 2 (Patients characteristics)

 What is the gene status of those patients? Is there any relationship between the gene status and PETCT features?

 Unfortunately, it is not possible to perform an analysis between genetic status and PET/CT features, since out of 36 patients only 28 have a genetic study, of which only 1 had a BRCA 1 mutation and 4 had variants of uncertain significance. The remaining 23 patients had no mutation in the BRCA genes.

The author should provide more information about the GLZLM-ZLNU

We have explained more about the definition of the feature and added a reference about their prognostic value on Hodgkin Lymphoma